

# A coupled wave-3D hydrodynamics model of the Taranto Sea (Italy): a multiple-nesting approach

Maria Gabriella Gaeta[1], Achilleas G. Samaras[1], Ivan Federico[2], Renata Archetti[3]

[1]CIRI-EC, Fluid Dynamics Unit, University of Bologna, Via del Lazzaretto 15/5, Bologna, 40131, Italy
[2]CMCC, Euro-Mediterranean Center on Climate Change, Via Augusto Imperatore 16, Lecce, 73100, Italy
[3]Department of Civil, Chemical, Environmental and Materials Engineering, University of Bologna, Viale Risorgimento 2, Bologna, 40136, Italy

*Correspondence to*: M.G. Gaeta (g.gaeta@unibo.it)

**Abstract.** The present work describes an operational strategy for the development of a multiscale modelling system, based on a multiple–nesting approach and open-source numerical models. The strategy was applied and validated for the Gulf of Taranto in South Italy, scaling large-scale oceanographic model results to high-resolution coupled wave -3D hydrodynamics simulations for the area of Mar Grande in Taranto Sea. The spatial and temporal high-resolution simulations were performed using the open-source TELEMAC suite, forced by wind data from the COSMO–ME database, boundary wave spectra from the RON Buoy at Crotone, and results from the Southern Adriatic Northern Ionian coastal Forecasting System (SANIFS) regarding sea levels and current fields. Model validation was carried out using data collected in the Mar Grande basin from a fixed monitoring station and during an oceanographic campaign in October 2014. The overall agreement between measurements and model results in terms of waves, sea levels, surface currents, circulation patterns and vertical velocity profiles is deemed to be satisfactory, and the methodology followed in the process can constitute a useful tool for both research and operational applications on the same field and as support of decisions for management and design of infrastructures.

## 1 Introduction

The ability to accurately represent hydrodynamics processes in the nearshore are essential for today's operational applications for coastal planning, decision support and assessment, since the majority of human activities are concentrated within the coastal zone. The estimate of sediment transport rates, the design of coastal defence structures and harbours, the forecast of pollutant concentrations, the regularization of ship routing, the knowledge of wave energy potential at specific locations are all among the applications requiring a complete reproduction of the wave-current induced processes.

High-resolution wave and hydrodynamics modelling offers an extensive range of capabilities regarding simulated conditions, works and practices, as well as providing with a wide array of data regarding nearshore hydrodynamics.



Nowadays, predictive operational oceanography takes into account regional, sub-regional and shelf-coastal scales, based on coupled models of wave, current and active tracer (sea temperature and salinity) dynamics, including interactions with the atmosphere by means of empirical bulk formulae. However, since the above mentioned oceanographic operational systems do not reach spatial resolutions lower than a few hundred meters, they are not able to take into account typical features of

coastal engineering scale: (i) non–linear processes of wave propagation, (ii) interactions with offshore and coastal structures, and (iii) boundary conditions induced by coastal processes, like river discharges (Sanchez-Arcilla et al., 2014). The combined effect of waves and currents in shallow waters was the subject of numerous studies since Longuet-Higgins and Stewart (1964) introduced the term of radiation stress in hydrodynamics equations, a net momentum flux induced by spatial variation of the wave action spectra responsible for phenomena like wave set-up, wave set-down and longshore currents.

Several authors have demonstrated that the coupling of wave and surge, tide (Holthuijesen, 2007; Roland et al., 2009; Wolf, 2009) and ocean currents (Hersbach and Bidlot, 2008; Benetazzo et al., 2013) is a key element to improve the accuracy of nearshore dynamics predictions. The most relevant aspects of these interactions define the influence of water-depth variation in modulating wave energy during propagation and breaking, and of tidal currents in steepening incident waves. Hence, correct wave forecasting in coastal waters is only possible through the implementation of a multiple‐nesting methodology,

based on the development of (i) numerical models at high spatial and temporal resolution, (ii) downscaling technique and (iii) a two-way coupling between hydrodynamics and spectral models.

Numerical models based on unstructured grids have the advantage of describing more accurately complicated bathymetry and irregular boundaries in shallow water areas and have been recently implemented to combine large-scale oceanic (regional) and small-scale coastal (local) dynamics (Lane et al., 2009; Liu and Xie, 2009; Xing et al., 2011; Guillou &

Chapalain, 2012; Ferrarin et al., 2013), showing a significant improvement in predicting wave field.

The availability of field measurements is essential, in the above context, in order to calibrate and validate the numerical models, thus enhancing their predictive capabilities (Scroccaro et al., 2004; De Serio et al., 2007).

In the following, the operational strategy for the development of a multiscale modelling system, based on a multiple-nesting approach and open-source numerical models, is presented and implemented focusing-at the highest resolution-on the Mar

Grande Area in the Sea of Taranto (South Italy). Section 2.2 describes the available field measurements, used to calibrate and validate the adopted numerical models, thus confirming the validity of the proposed approach. Simulation results of 2D wave-current interactions for the entire Gulf of Taranto are reported in Section 3 and provide the offshore boundary conditions in terms of wave forcings for the small-scale 3D model of the Taranto Sea, presented in Section 4. The computed current pattern developing in Mar Grande and the evaluation of flux exchanges with the open sea are discussed in

comparison to the collected field data.





## 2 Materials and methods

### 2.1 Multiple-nesting approach and model description

The proposed multiple-nesting approach is represented in Fig. 1 where the computational domains and bathymetries of the different models composing the numerical chain are shown. The methodology is based on four levels of downscaling as described in the following.

(a) The large-scale system for the entire Mediterranean basin (MFS, Mediterranean Forecasting System, e.g. Pinardi et al., 2003; Pinardi and Coppini, 2010), the first model of the chain (Fig. 1a). It is developed and operationally maintained by the National Institute of Geophysics and Vulcanology (INGV) providing operational forecasting products in the Copernicus Marine Environment Monitoring Service (http://marine.copernicus.eu/). The current MFS implementation is based on the NEMO (Nucleus for European Modelling of the Ocean; Madec et al., 1998) finite-difference code with a horizontal resolution of 1/16° (7 km approximately) and 72 unevenly spaced vertical levels. The forecasting system is provided by a data assimilation system based on the 3DVAR scheme developed by Dobricic and Pinardi (2008).

(b) The Southern Adriatic Northern Ionian coastal Forecasting System (SANIFS, in Fig. 1b), a coastal-ocean operational system providing short-term forecasts and built on the unstructured-grid finite-element three-dimensional hydrodynamic SHYFEM model (Umgiesser et al., 2004; Ferrarin et al., 2013). SANIFS resolution ranges from 3 km in open-sea to 500-50 m in coastal areas. The model configuration has been outlined to provide reliable hydrodynamics and active tracer forecasts in open-sea and shelf-coastal waters of Southeastern Italy (Apulia, Basilicata and Calabria regions) thanks to its high-properly adapted-horizontal resolution. The model is forced: (i) at the two lateral open boundaries through a full nesting strategy directly with the MFS (temperature, salinity, sea surface height and currents) and OTPS (tidal forcing) fields; and (ii) at the sea surface through two alternative atmospheric forcing datasets (ECMWF-12 km and COSMOME-6 km) via MFS-bulk-formulae (e.g. Pinardi et al., 2003; Castellari et al., 1999). SANIFS features have been validated in Federico et al. (2016) comparing model results with observed data.

(c) The coupled wave-2D hydrodynamics model of the Gulf of Taranto (Fig. 1c), based on the respective modules of the TELEMAC suite (Benoit et al., 1996). In its setup, waves are being propagated from the Crotone buoy using the spectral module TOMAWAC (henceforth denoted as TOM) and wave-induced currents are reproduced using the 2D hydrodynamics module TELEMAC2D (henceforth denoted as TEL2D). The model is forced by SANIFS results (Fig. 1b) as initial/boundary conditions at both the offshore field boundary and the free surface.

(d) The coupled wave-3D hydrodynamics model of the Taranto Sea (Fig. 1d), also based on the respective modules of the TELEMAC suite (Benoit et al., 1996). For the spectral module, the imposed waves at the offshore boundary are extracted from the 2D model of the Gulf of Taranto (Fig. 1c); for the hydrodynamics module TELEMAC3D (henceforth denoted as TEL3D), the imposed 3D offshore boundary and surface conditions, as well as the 3D initial conditions, are extracted from SANIFS simulations (Fig. 1b).



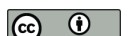

Table 1 shows the numerical features of the proposed nesting approach: the simulated period, the mesh-size range and the spin-up time are reported for the different scale models. The spin-up time, i.e. the initial run-time needed in order for the models to arrive to relative stability regarding hydrodynamics, decreases with the increase in spatial and temporal resolution. The development of the very high-resolution models (Fig. 1 c and d) of the proposed downscaling chain is the main object of

the present study, as the overall operational strategy. The TELEMAC suite is distributed under a General Public License (GPL) and is available at TELEMAC (2015). It is a finite element based solver for shallow water flows, wind wave propagation, ground water flows, tracer transport, sediment transport and morphodynamics. The parallel version of the last released suite was used on GALILEO, a CINECA machine characterized by a Linux Infiniband Cluster architecture with 8,256 nodes, operating under the EU-Innovative SuperComputing Research Allocation (ISCRA) program.

In the proposed approach, the wave and 2D/3D hydrodynamics modules of the TELEMAC system are implemented in order to propagate offshore waves and currents and reproduce nearshore dynamics.

TOM is a third-generation spectral wave model, solving a simplified equation for the spectro-angular density of wave action by means of a finite-element type method, in order to describe wave propagation towards coastal areas.

TEL2D simulates 2D hydrodynamics by solving the depth-averaged flow equations (proposed by de Saint-Venant (1871) in

order to study free surface hydraulics and tracer transport for both transient and permanent conditions. The space is discretized by a series of Delauney triangular unstructured elements. Source terms in dynamic equations are modelled to represent the Coriolis force, bottom friction and wind action; active (temperature and salinity) and passive tracers can be also reproduced, as source or sink terms.

Coupled wave-2D hydrodynamics modelling was discussed in Samaras et al. (2016), based on the intercomparison of the

TELEMAC suite and the well-known commercial software MIKE21 by ©DHI Group.

TEL3D (Hervouet, 2007) solves the three-dimensional Navier-Stokes equations, with the option of non-hydrostatic pressure hypothesis, and includes: (i) the use of a finite element unstructured grid which allows selective refinement of the mesh at key locations in the domain and boundary fitting method for vertical discretization; (ii) the transport-diffusion equations of intrinsic quantities (temperature, salinity, concentration), in order to reproduce 3D hydrodynamics including the transport of

active and passive tracers; (iii) a wide range of options for vertical turbulence modelling. The numerical solver used in the model is based on a fractional step technique in which the governing equations are split into fractional steps and treated using appropriate algorithms for the advection and diffusion of flow variables. The advection of velocities and water elevations is done with the semi-implicit Streamline Upwind Petrov-Galerkin scheme, and the conjugate gradient method is used to solve the diffusive terms.

When using the same horizontal discretization, the modules TEL2D and TEL3D can be directly coupled (two-way coupling) to the spectral module TOM in order to reproduce the dynamics of wave-driven currents: the gradients of the radiation stress induced by waves are computed using the theory of Longuet-Higgins and Steward (1964) as part of the hydrodynamics equations.



In order to implement the proposed multiple-nesting approach, the authors properly modified each of the aforementioned modules in order for them to be able to read: (i) space-varying initial conditions for 2D/3D currents and water elevations (in TEL2D/TEL3D) and 3D environmental fields (in TEL3D, i.e. temperature and salinity); and (ii) time-/space- varying boundary conditions of wave spectra (in TOM), 2D/3D currents and water elevations (in TEL2D/TEL3D), and 3D

environmental fields (in TEL3D) and (iii) time-space varying surface conditions for wind (both in TOM and TEL2D/TEL3D) and meteo-climatic conditions (in TEL3D, i.e. atmospheric pressure, air humidity, cloud cover, etc.).

Blue Kenue©, a freeware tool developed by the National Research Council of Canada, is used to prepare the variable-density triangular meshes of the study area and to visualize/process models results.

The setup of TELEMAC applications consists of 6 sequential steps (to be applied for each module), namely: mesh

generation, adaptation of a steering file, extraction of initial/boundary conditions from larger-scale models, scripting and compiling modules' subroutines in FORTRAN, and post-processing.

## 2.2 Description of the investigated area and field data collection

The studied area (Fig. 1c), located in the Ionian Sea, is scientifically interesting due to its connection with the Adriatic Sea system, inducing the large-scale circulation structure known as the Taranto anti-cyclonic Gyre, as well as other minor

cyclonic structures near the shelf. In addition, the area is considered to be a vulnerable and sensitive area, affected by chemical and biological pollutant industrial discharges and intense ship traffic (Di Leo et al., 2013). Accordingly, the accurate  representation of its circulation regime is essential for the reconstruction of mixing and dispersion processes in the area as well (Mossa, 2006).

Figure 2 shows the study area of Taranto Sea, comprising Mar Grande, Mar Piccolo and the adjacent industrial area (see also

Fig. 1d). The semi enclosed basin of Mar Grande is delimited by Punta Rondinella, S. Paolo Island and Capo S. Vito and is directly connected to the Gulf of Taranto through one large opening at its south boundary and two smaller ones at the western one; Mar Grande host the commercial and military harbours of the city of Taranto, as well as a number of mussel farms close to Capo S. Vito. Mar Piccolo is divided in two basins, First Seno and Second Seno; it is characterized by the presence of a number of submarine fresh water springs, with a combine discharge of around 5 $m^3$/s (De Pascalis et al., 2015)

and by numerous mussel farms; it is connected to Mar Grande through two navigable openings, named Porta Napoli and Canale Navigabile. A submarine freshwater source with an averaged discharge of 17 $m^3$/s is also present at the mouth of Porta Napoli (De Serio and Mossa, 2015).

The bathymetry and shoreline used in the present work resulted from the digitization of nautical charts acquired from the Italian National Hydrographic Military Service. Since the areas of Mar Grande and Mar Piccolo are of particular interest for

their hydrodynamic regime and environmental fragility, they have been the place for a series of short- and long- term monitoring activities over the years; the most recent ones are listed in the following, their characteristics also reported in Fig. 2.



- The monitoring station in Mar Grande-MEDA - (De Serio and Mossa, 2015) is located at 17° 12.9' E and 40° 27.6' N, in the middle of the Mar Grande basin (black circle in Fig. 2) and was installed in February 2014 under the Italian projects PON n. H51D11000050007 and RITMARE. The station consists of one upward-looking Acoustic Doppler Current Profiler (ADCP) placed on the sea bottom, at a depth of -22.65 m. The instrument is able to provide, with a frequency of 30 min, information on: (i) waves, in terms of significant height, peak and mean direction and peak period, (ii) currents and (iii) free surface elevations. The station also has instrumentation to record environmental variables of the atmosphere (i.e. wind speed and direction, air temperature and humidity, atmospheric pressure, net solar radiation) and sea (i.e. water temperature, pressure, conductibility, dissolved oxygen concentration).

- A RMN gauge of the Italian Mareographic Network is located at 17° 13' 25.55" E and 40° 28' 32.17" N, close to the Porta Napoli channel entrance, and collects at a 10 min interval measurements of wind speed and direction, sea water level and mean water temperature, air temperature, atmospheric pressure and humidity and cloud cover.

In 2014, an oceanographic cruise campaign (Pinardi et al., 2016) named MREA2014 (Marine Rapid Environmental Assessment) was performed in the area of the Gulf of Taranto during the period from 1 to 10 October, by using Italian Navy Hydrographic Institute cruise-ship to collect environmental data at three scales: large, shelf-coastal scale and coastal-harbour scale. At this latter one, which is of interest in this work:

- ADCP transects (green lines in Fig. 2) were carried out at 5 different sections (Punta Rondinella, Western Opening, Canale Navigabile and Porta Napoli and between the points of S. Paolo Island and Capo S. Vito) to estimate surface current patterns and circulation;

- 3 groups of 5 drifters (purple stars in Fig. 2) were released in the period from 5 October 2014 at 10:40 am to 6 October 2014 at 1:00 pm, in order to estimate the surface currents (up to -2 m of water depth);

- 31 CTD profiles (black circles in Fig. 2) were collected on 5 October 2014 to measure water temperature and salinity profiles.

The drifter trajectories and ADCP mean velocities will be compared with the numerical results and the reproduced patterns of surface velocities, including secondary circulation inside the basin, will be discussed in Section 4.

## 3 Coupled wave -2D hydrodynamics model of the Gulf of Taranto

### 3.1 Model set-up

The computational field for the Gulf of Taranto (Fig. 1c) was discretized using a variable density unstructured mesh in Blue Kenue©. Close to the shoreline, the reproduced mesh has not a detailed resolution since the purpose of the 2D model is mainly to propagate waves from offshore and extract wave field at the offshore boundary of the 3D model; hence, harbour infrastructures, breakwaters and small islands are not represented in the area discretization.



TEL2D and TOM are implemented on a domain extending on the whole Gulf of Taranto from 16° 28' E to 18° 19' E and 39° 23' N to 40° 31' N. The unstructured mesh comprises 91,750 nodes and 179,000 finite elements with a size of 1 km offshore to 50 m close to the -10 m iso-depth. All the 2D runs are carried out for the period 1-10 October 2014.

TOM is driven by the wave components extracted every 30 min from the offshore buoy located in Crotone. The wave spectra are discretized to 16 directions, with a band width of 24°. Non-linear energy conservative processes are included in the model set-up according to the authors' experience reported in Samaras et al. (2016). Wind-generated spectral energy source is modelled under the WAM-cycle 4 model (Komen et al, 1984; Komen et al., 1994); the atmospheric forcing datasets from COSMO-ME (based on the COSMO model; Baldauf et al., 2011, among others) are adopted in the TELEMAC models, properly extrapolated to reconstruct wind field in the zone between the last wet mesh nodes of the high resolution model and the actual coastline following Kara et al. (2007). An interpolation procedure is used to create the oceanic fields over the TELEMAC grid, as described by De Dominicis et al. (2013), and space-time varying wind conditions are imposed in the models.

Figure 3 shows a comparison at the Mar Grande and RMN stations of the wind velocity magnitude (top panels) and direction (bottom panels) between the measurements (red squares) and COSMO-ME values (blue lines) interpolated over the TELEMAC grid. The agreement is quite good, with the exception of the maximum event corresponding to the higher wave. This is attributed to the fact that large-scale atmospheric model as COSMO-ME does not properly reproduce local atmospheric events, main responsible of fast perturbation, as the one occurring during the night between 6 and 7 October.

A model calibration is performed varying the values of wind drag coefficient $C_D$ and the bottom friction included in the model in order to achieve a good agreement with the measurements in terms of depth-average velocities, free surface elevation and wave height. The final set-up of TEL2D includes a wind drag coefficient $C_D$ equal to $3.5 \times 10^{-3}$ in order to compensate the underestimation of the actual wind by the COSMO-ME fields. The bottom friction is computed according to Chezy's law, with a Gauckler-Strickler coefficient equal to 40 $m^{1/3}/s$; a constant Coriolis coefficient $f$ is set equal to $1.101 \times 10^{-4}$ N/m. The Smagorinsky model (Smagorinksy, 1963; Hervouet, 2007) is chosen to represent the horizontal turbulent dissipation, with a velocity diffusivity coefficient equal to 0.1 $m^2/s$.

Wave-driven radiation stresses computed in TOM are fed into the hydrodynamics module TEL2D, directly coupled in order to reproduce wave-driven currents. Initial conditions of water elevations and depth-averaged currents are extracted from the SANIFS model and interpolated over the 2D TELEMAC grid, imposing the no-slip conditions on the coastline. TEL2D is driven by the free-surface elevations and the depth-averaged currents (extracted hourly by SANIFS) and by wind forcings varying in time and space. A spin-up time of 2 days (1-2 October 2014) was necessary to the coupled model to get stable conditions for currents.

## 3.2 Model results

Numerical results from the 2D hydrodynamics module TEL2D coupled to the spectral module TOM are obtained in order to force the 3D model of the Taranto Sea with the waves propagated from Crotone to the shelf-coastal area. In order to validate

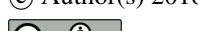



the results, the time series of free surface elevation and significant wave height are compared to the data collected at the Mar Grande station. The simulated period was characterized by small waves in Mar Grande until the day 5 October 2014 when wind increased and changed in direction (also shown in Fig. 3).

TEL2D managed to capture well tidal the tidal constituents (Fig. 4) both in amplitude and frequency in comparison to the observations. The effects of tide-induced water elevations and currents on wave height are analyzed in the area of the Gulf of Taranto, where waves measured at the Crotone buoy are propagated from the offshore boundary (Fig. 5). The coupled TEL2D+TOM model predicts satisfactorily time series of wave height in comparison with measurements at the Mar Grande station, while wave climate is roughly underestimated with using the standalone TOM run, as described in Fig. 6. The 2D model results are in accordance to the field evidence reported in De Serio and Mossa (2015) who showed how tidal flow induces steepening of the incident wave fields, thus increasing the wave height. Numerical results tend to slightly overestimate the wave height after the peak (where $H_s$ equal to 0.4 m) characterizing the storm between 6 and 7 October 2014: wind directions rapidly changed during this short event that the imposed atmospheric forcings from COSMO-ME don't reproduce.

## 4 Coupled wave-3D hydrodynamics model of the Taranto Sea

### 4.1 Model set-up

The model of the Taranto Sea is implemented on a domain extending from 17° 7' E to 17°12' E and from 40° 27' N to 40° 29' N (Fig. 7, top panel). The study area includes Mar Grande (maximum depth = -50 m), with all the islands and the coastal defence structures properly represented in the constructed mesh, including the harbours of the city of Taranto (Fig. 7, bottom panels) and the western open sea, in front of the industrial area of the city, and Mar Piccolo, with the two small basins.

The horizontal unstructured mesh is constructed adopting the following mesh-size rules, with an edge growing ratio of 1.2: 100 m at the offshore boundary, where each node corresponds to a node of the SANIFS mesh, 50 m inside the liquid domain, 10 m at the coastline, harbour structures and islands. 3D space is represented as a series of 2D triangular unstructured meshes with 55,399 nodes and 105,751 elements, at defined levels between the bottom and the free surface. The implemented vertical discretization follows a 17 z-level scheme, following the set-up adopted in SANIFS, where the thickness of each layer is equal to 2 m.

In TEL3D, three fractional steps are implemented to solve the governing Navier-Stokes equations, consisting in (i) finding out the advected velocity components by only solving the advection terms in the momentum equations; (ii) computing, from the advected velocities, the new velocity components taking into account the diffusion terms and the source term in the momentum equations; (iii) computing the water depth from the vertical integration of the continuity equation and the momentum equations only including the pressure-continuity. In the present study, horizontal turbulence is modelled using the sub-grid Smagorinsky scheme (Smagorinksy, 1963; Hervouet, 2007); at the vertical scale, the mixing-length turbulence





closure is applied following the classical formulation by Prandtl (1925). The values of bottom friction, wind drag coefficient and Coriolis coefficient used in TEL3D are equal to the ones estimated for the 2D simulations of the Gulf of Taranto.

Water temperature $T$ and salinity $S$ are modelled as active tracers, following the mass-balance equation; their values at each time step and node are used to compute the water density $\rho$ variation according to the state equation (Hervouet, 2007), as:

$$\rho = \rho_0 \left[ 1 - \left[ 7 \cdot \left( T - T_0 \right)^2 - 750 \cdot S \right] \cdot 10^{-6} \right] \quad\quad\quad (1)$$

where $\rho_0$ is the reference water density equal to 1025 kg/m$^3$, $T_0$ is the reference temperature equal to 4°C. The density variation of the water is therefore estimated following Eq. (1) and included in the pressure continuity.

Waves resulting from the coupled TEL2D+TOM run of the entire Gulf of Taranto (Section 3) are imposed on the offshore boundary of the 3D model. The subroutine *limwac.f*, defining the imposed spectral energy conditions at the boundaries, was

properly modified in order to get at each node of the offshore boundary and at each time step of the run the values of significant wave height, peak period and mean direction, extracted and linearly interpolated in time by the 2D simulation results. Figure 8 (left panel) shows the colormap of the imposed significant wave height $H_s$, where x-axis reports the time, y-axis the relative distance on the offshore boundary. The prescribed signal variation on the boundary outline is extracted for the exemplary instance of 6 October 2014 at 2:00 pm (noted as a dotted line in the left panel) and shows in the right panel.

In analogy to the 2D model, wind data from COSMO-ME are also imposed at surface, modifying the subroutine *venuti.f* in order to have varying in time and space forcings.

The spectral module TOM is coupled with the 3D hydrodynamics model TEL3D, properly modified in order to account for time-space varying: (i) water elevations, velocities and active tracers (water temperature and salinity) along the offshore boundary (in the subroutine *bord3d.f*); and (ii) wind over the entire domain (in the subroutine *meteo.f*). Initial conditions of

water elevations, velocities and tracers are extracted from SANIFS results at the day 3 October 2014 and are extrapolated and interpolated at each node of the domain and at each horizontal layer, modifying the subroutine *condin.f* and adopting the procedure described in De Dominicis et al. (2013) to impose the data on the TELEMAC grid. A spin-up time of 2 days (3- 4 October 2014) was necessary to the coupled model to get stable conditions for both currents and tracers.

The three most significant waste-water discharges in the study area (De Pascalis et al., 2015) were also introduced/prescribed

as boundary conditions in the model setup: the ILVA water pump withdrawal with yearly-averaged discharge of about 27 m$^3$/s from Mar Piccolo, and the two ILVA wastewater releases, northern Punta Rondinella and outside Mar Grande, with a total yearly-averaged discharge of 26 m$^3$/s. The contribution of submarine fresh-water sources inside Mar Piccolo was neglected in the present study.

## 4.2 Model results

The capability of the proposed multiple-nesting approach and of the implemented models is investigated on the reproduced dynamics of Mar Grande, focusing, in particular, on current trajectories and intensity. Figure 9 shows the surface velocity field in the area of Mar Grande: the values represent the first two layers' depth-averaged (up to -2 m of water depth) of the



3D velocity results in the interval from 10:00 am to 12:00 pm of the day 5 October 2014, in order to compare these numerical outputs with the average observed trajectories of each drifter group (black arrows in the figure).

The current pattern in the area where the drifters were released is adequately captured by the coupled model: the average objects' propagation is estimated to be directed towards around 270° N, except for the drifter group D2, moving northwards, while the computed surface velocity is slightly underestimated at the area close to the release point of the drifter group D1. Figure 10 shows the depth-averaged velocities along 4 opening sections (green lines in Fig. 2; the ADCP data collected along the southern opening were neglected due to noisy signals for the severe climate conditions during the measurements) where the ADCP collected current profiles in the measurement interval from 7:20 am to 9:30 am of the day 5 October 2014. An excellent agreement between observations and the numerical results is observed for the mean flux between Mar Grande and Mar Piccolo (along Porta Napoli and Canale Navigabile sections) and the western open sea (along Punta Rondinella and West Opening sections). The counter-clockwise circulation at the surface in Mar Grande is also in accordance to the comparison at the ADCP transects as reported in Table 2: a negative mean flux exiting the Mar Grande basin is observed both in measurements and computation; discharges along the sections are also well reproduced by the model, along with the mean flux velocity and direction.

The time series of currents and the vertical profiles at the point of the Mar Grande station are presented in Figs 11 and 12, respectively: in both figures, the numerical results from the TEL3D+TOM run are compared with the observations and the SANIFS outputs. The overall agreement of computations is quite good, with a mean error in the depth-averaged velocity magnitude equal to 24% and 33% for TEL3D+TOM and SANIFS runs, respectively. Vertical profiles of velocity at the Mar Grande station are also well captured by the model: the daily-averaged values in the period from 5 to 7 October 2014 show a good agreement with observations. The combined effects of: (i) the coupling of hydrodynamics and wave dynamics; (ii) the inclusion of Mar Piccolo dynamics in the modelling system, (iii) the introduction of the main wastewater discharges; and (iv) the increase in spatial and temporal resolution, have led to a better representation of the circulation regime inside Mar Grande by the proposed modelling approach.

In general, the consistent trends of a motion field and circulation is well predicted and reproduced: on the contrary respects to the Gulf of Taranto, where the current circulation is highly influenced by the local topography and is characterized by an anti-cyclonic trend (Federico et al., 2016), directed towards the south-east, the flow becomes slower where a rough bathymetry occurs inside Mar Grande, resulting into an opposite trend with the development of a small cyclonic pattern at the surface.

## 5 Conclusions

The present work proposes a multiple-nesting methodology for the assessment of nearshore wave dynamics and hydrodynamics, based on a downscaling approach-from oceanographic to coastal engineering scale-and the implementation of open-source numerical models, customized for specific user purposes and accounting for wave-hydrodynamics processes.





The approach, deemed to be useful for environmental planning and decision support systems for maritime hazards, was applied to scale down from the Mediterranean Forecasting System-MFS (regional system), to the Southern Adriatic Northern Ionian coastal Forecasting System-SANIFS (mesoscale-shelf-coastal), to very high-resolution applications of the models comprised in the TELEMAC suite.

The aforementioned methodology was applied to the case study of the Taranto Sea, South Italy, a vulnerable and sensitive area including Mar Grande and Mar Piccolo and the adjacent industrial area. Field measurements, available from a recent oceanographic campaign (MREA campaign, October 2014) and monitoring stations in Mar Grande, were used to calibrate and validate the TELEMAC models. Nesting regional and local systems based on the coupling of wave propagation and circulation modules have been used to propagate waves from the offshore boundary of the Gulf of Taranto to the nearshore.

2D simulations of wave-current interactions were implemented through the two-way direct coupling of the spectral module TOMAWAC and hydrodynamics module TELEMAC2D. Models were forced at the offshore boundary by: (i) wave spectra from the Crotone buoy, and (ii) free surface elevation and currents extracted from SANIFS; (iii) at the surface by wind data from the COSMO-ME model. The computed time series of sea levels and wave heights show a good agreement with measurements carried out in Mar Grande, revealing that the inclusion of tide-induced time-/space- varying water elevations

and velocities in wave modelling is essential for an accurate representation of the wave field during storms.

A coupled wave-3D hydrodynamics model of the Taranto Sea was applied to investigate the dynamics of the area of Mar Grande, focusing, in particular, on current trajectories and intensity. The consistent trends of the current field and circulation are well reproduced with the development of a small cyclonic pattern at the surface, in accordance to the observed trajectories of the released drifters. The fluxes of water-masses between the open sea and Mar Grande and Mar Piccolo are

well reproduced by the model too, showing a good agreement with the ADCP data. The intercomparison with SANIFS results in terms of current profiles at Mar Grande station shows an overall satisfactory agreement with the observations and is deemed to provide useful insights on implemented model features.

The results highlight the capabilities of the proposed multiple-nesting approach to comply with the observed sea conditions at coastal-harbour scale, considered to be essential for a wide range of coastal engineering and management purposes.

Overall, the methodology is deemed to be of general interest for ocean and coastal modellers involved in: (i) the development of procedures for nesting high-resolution models to general circulation models of regional/sub-regional scale (as well as their validation based on field data); (ii) the adaptation of coastal models to platforms of operational oceanography; (iii) the implementation of the above to coastal planning and design (coastal zones and harbours).

**Acknowledgments**

This work was performed and funded in the framework of the Italian Flagship Project "TESSA - Development of Technologies for the Situational Sea Awareness" supported by the PON01_02823/2 "Ricerca & Competitività 2007-2013" program of the Italian Ministry for Education, University and Research. The authors would like to thank Prof. Michele



Mossa from the Technical University of Bari and CNR-ISMAR for proving field data at the Mar Grande station and from MREA2014 campaign respectively. The Italian Flagship Project RITMARE (SP3-WP4-AZ2) is also acknowledged.

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



Table 1. Features of the multiple-nesting approach: simulated period, mesh-size and spin-up time for MFS (Fig. 1a), SANIFS (Fig. 1b), TEL2D+TOM (Fig. 1c) and TEL3D+TOM (Fig. 1d) models.

| Models | Simulated Period | Mesh-Size Range (m) | Spin-up time (days) |
|---|---|---|---|
| MFS[a] | - | 6000-7000 | - |
| SANIFS | 30/09-10/10/2014 | 50-3000 | 3 |
| TEL2D+TOM | 1-10/10/2014 | 50-1000 | 2 |
| TEL3D+TOM | 3-7/10/2014 | 10-100 | 2 |

5   [a] MFS runs in operational mode

Table 2. Water-mass exiting Mar Grande on 5 October 2014: comparison between numerical results and field data from ADCP transects.

| Section Name | Numerical results/ADCP data | | |
|---|---|---|---|
| | Mean velocity (cm/s) | Mean flux direction (°N) | Discharge (m$^3$/s) |
| Porta Napoli | 9/9.5 | 15/11 | 100/95 |
| Canale Navigabile | 30/34 | 23/17 | 220/195 |
| Punta Rondinella | 27/30 | -50/-16 | 170/200 |
| Western Opening | 18/21 | -30/-16 | 17/19 |



Figure 1. Multiple-nesting approach from the Mediterranean Sea scale with (a) MFS and (b) SANIFS to coastal engineering scale model performed with the TELEMAC suite (c) in 2D for the Gulf of Taranto and (d) in 3D for the Taranto Sea: maps show the adopted bathymetry.





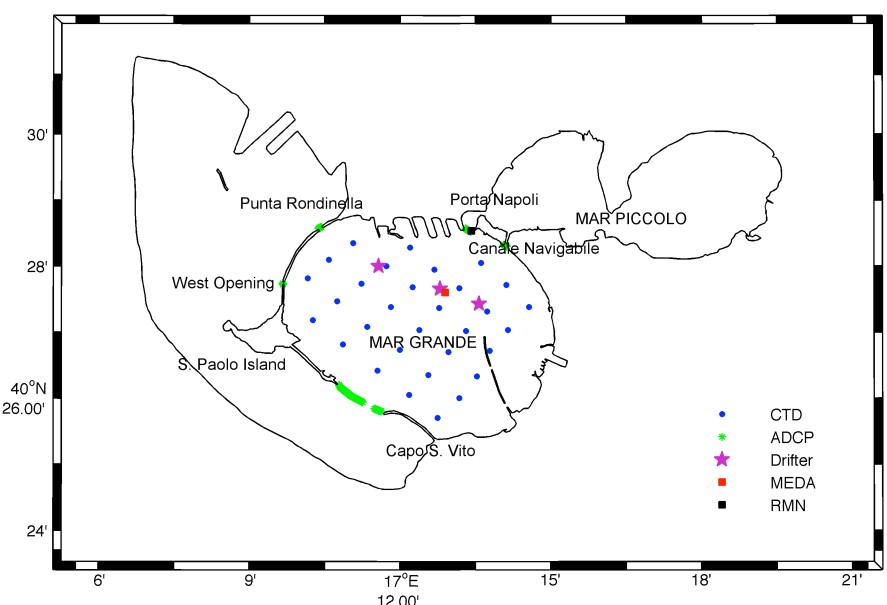

Figure 2. Map of the Mar Grande study area and points/locations of available measurement data sets from MREA14 campaign (CTD, ADCP, and drifters), the Italian Mareographic Network (RMN) and the Mar Grande station (MEDA).



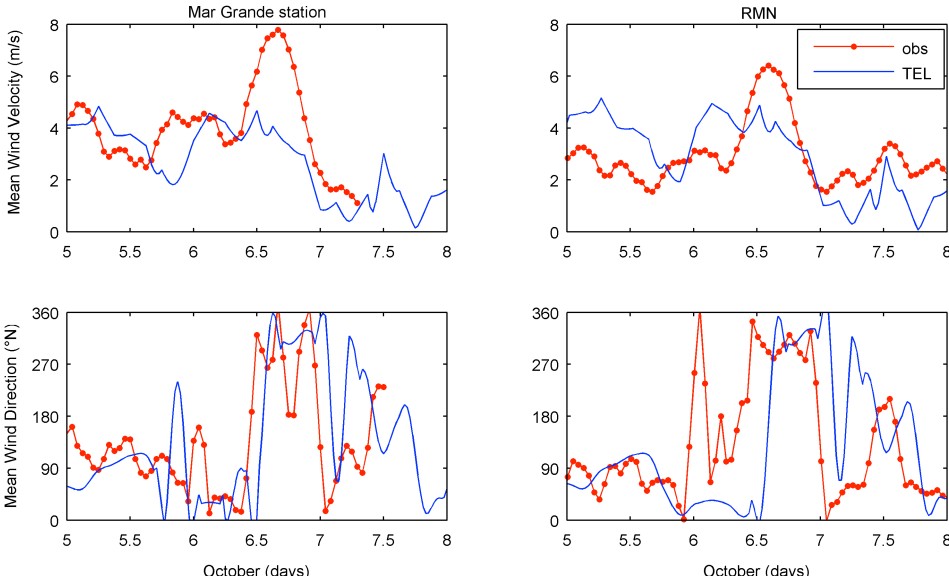

Figure 3. Wind characteristics at Mar Grande (left panels) and RMN (right panels) stations: velocity magnitude (top panels) and direction (bottom panels) from the interpolated-COSMO-ME values on the TELEMAC grid (blue line) and from observations (red line, circle markers).




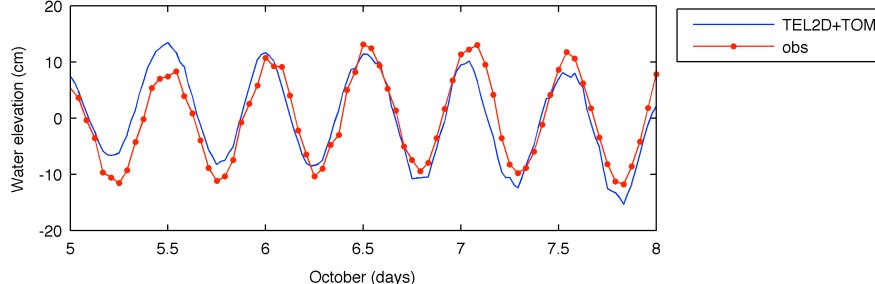

Figure 4. Time series of the tidal water elevation at the Mar Grande station: numerical results from the coupled TEL2D+TOM run (blue line) and observations (red circles).





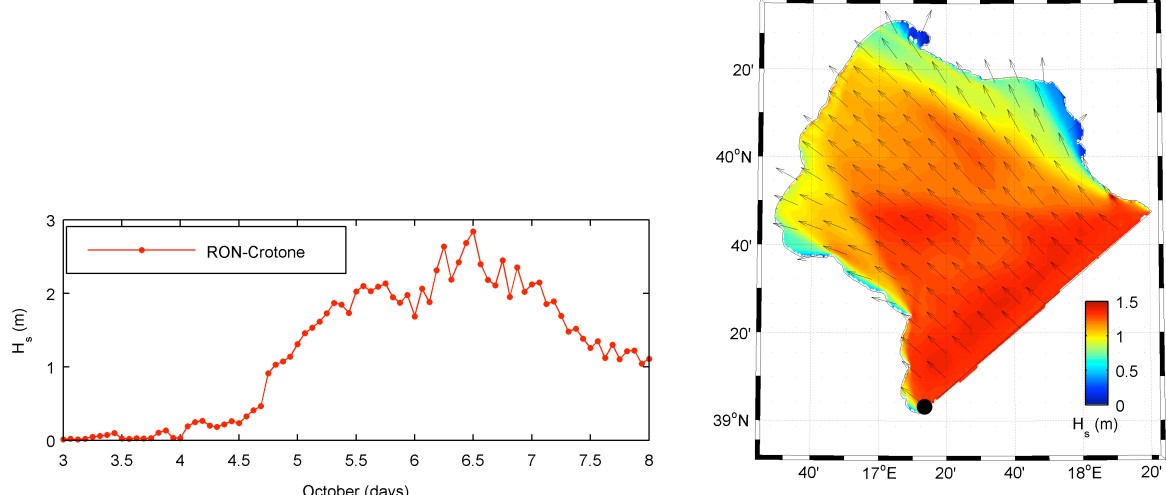

Figure 5. Numerical results from the TEL2D+TOM run for the Gulf of Taranto: significant wave height from Crotone buoy (black circle) imposed on the offshore model boundary (left panel), and snapshot (right panel) of wave propagation on 6 October 2014 at 2:00 pm (colormap of the significant height $H_s$ and arrows of the mean direction).





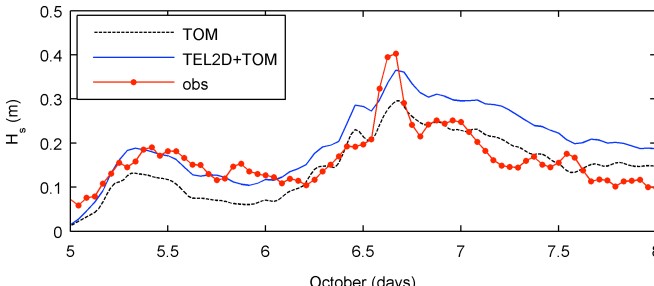

Figure 6. Significant wave height $H_s$ at the Mar Grande station: numerical results from the standalone TOM run (black line), the coupled TEL2D+TOM run (blue line) and observations (red circles).





Figure 7. 3D model of the Taranto Sea: the overall grid (top panel) and two detailed zones (bottom panels) close to the Commercial Harbor (a) and the Military Harbor (b) of Taranto; background images are from Google Earth, 2016, privately processed.





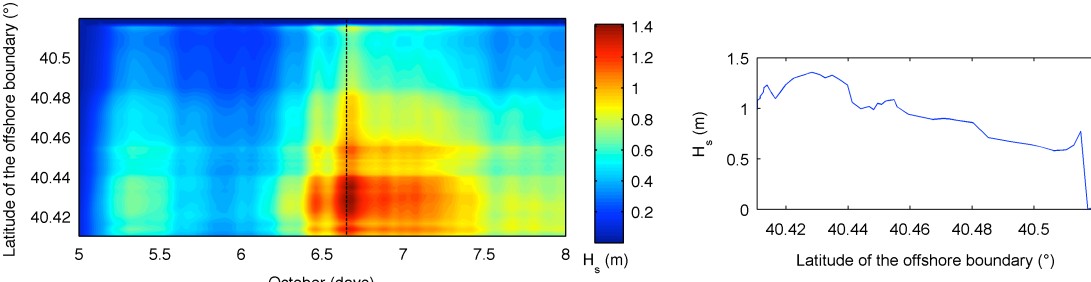

Figure 8. Colormap of the significant wave height $H_s$ imposed on the offshore boundary of the 3D model in time (left panel) and prescribed signal variation on the boundary outline extracted for an exemplary instance (right panel): instance of 6 October 2014 2:00 pm noted as a dotted line in the left panel.





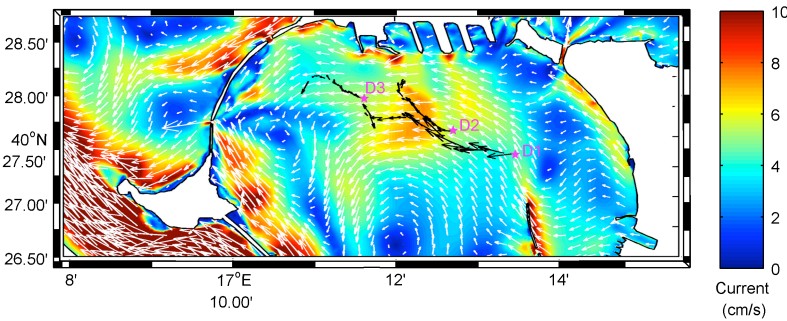

Figure 9. Surface velocity in the area of Mar Grande on 5 October 2014 (average over 10:00 am-12:00 pm): numerical results from coupled TEL3D+TOM (colormap for the velocity magnitude and white arrows for directions) and drifter routes (in black arrows).




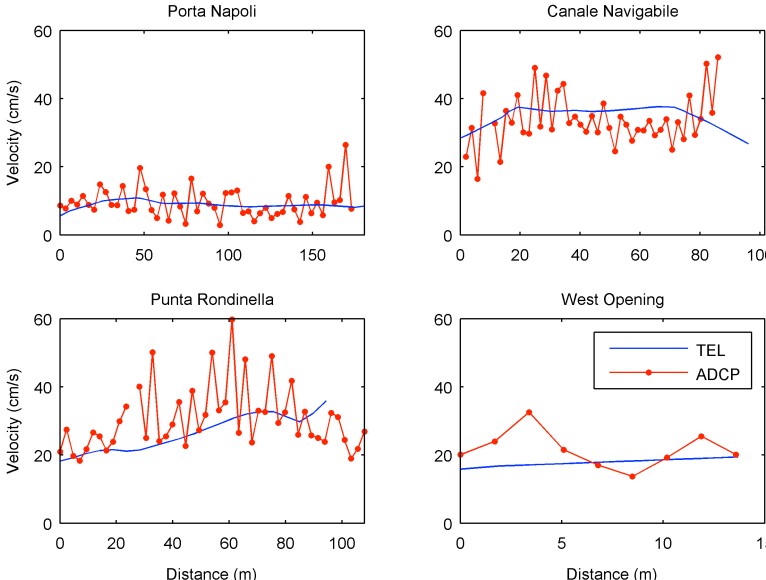

Figure 10. Depth-averaged velocity along 4 opening sections in Mar Grande: TEL3D+TOM results (blue line) and ADCP measurements (red line-circle markers).





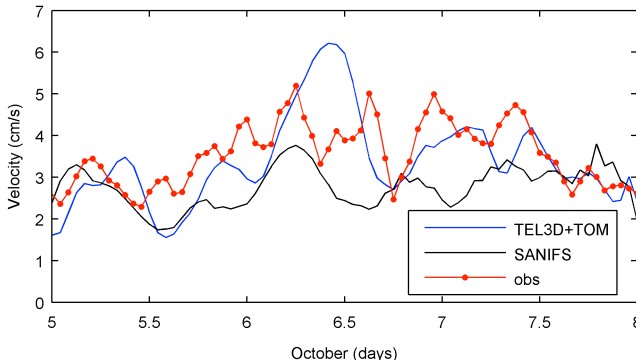

Figure 11. Depth-averaged velocity at the Mar Grande station: numerical results from the coupled TEL3D+TOM run (red line), SANIFS run (black line) and observations (red lines - circle markers).





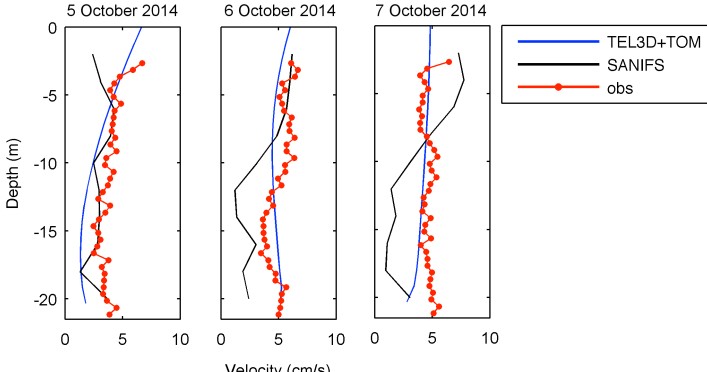

Figure 12. Daily-averaged vertical profile of current at the Mar Grande station: numerical results from the coupled TEL3D+TOM (blue line), SANIFS (black lines) and observations (red line-circle markers).

