# Peer review of "A coupled wave-3D hydrodynamics model of the Taranto Sea (Italy): a multiple-nesting approach"

_Natural Hazards and Earth System Sciences, 2016_

## Referee Comment (RC1) · Anonymous Referee #1 · 14 Apr 2016

Very interesting work, including different model coupling and validation with field measurements.

I recommend its publication.

Comments: -In Fig. 9 a reference vector would help more than the color scale. -Also in the area of Mar Grande (i.e. Figure 9 area) a figure with significant wave height distribution could be added.

---

## Referee Comment (RC2) · Anonymous Referee #2 · 18 Apr 2016

The manuscript describes a set of numerical applications finalized to reproduce the main hydrodynamics in a coastal area of the Mediterranean Sea, the Gulf of Taranto. Multiple nesting techniques and different numerical tools are applied to provide operational predictions of waves, currents and water levels for an area characterized by a high risk of pollution impacts. The paper falls within the scope of the journal. The arguments treated are very interesting and promising, nevertheless a major revision of the manuscript is required before being published.

Main concerns refer to the overall manuscript structure and to the quantification of the model accuracy. As a first, I suggest to separate the methodological sections from the results sections. In the present version the sections 3.1 and 4.1 describing the different models setups are included into that part of the paper should be dedicated to the description and discussion of the model results.

[Figure]

The Section 2.1 could be split into Model Description (2.1) and Model setup or Nesting Procedure or similar (2.2) including all the informations (including the Sections 4.1 and 4.3) needed to understand the different numerical experiments you performed. The Section 2.2 should describe only the field data, a specific Site Description Section could be included after the Introduction and before the methods.

Regarding the content of the methodological sections, you have to strongly reduce and try to clarify them. In particular, it is not necessary to describe the modifications you carried out to the model codes e.g. page 5 " In order to implement the proposed multiple-nesting approach, the authors properly modified each of the aforementioned modules in order for them to be able to read: (i) space-varying initial conditions for 2D/3D currents and water elevations (in TEL2D/TEL3D) and 3D environmental fields (in TEL3D, i.e. temperature and salinity); and (ii) time-/space- varying boundary conditions of wave spectra (in TOM), 2D/3D currents and water elevations (in TEL2D/TEL3D), and 3D 5 environmental fields (in TEL3D) and (iii) time-space varying surface conditions for wind (both in TOM and TEL2D/TEL3D) and meteo-climatic conditions (in TEL3D, i.e. atmospheric pressure, air humidity, cloud cover, etc.). " This can be deleted it is not relevant to the scope of a research paper. "

Again, it is not necessary the author mentions and describes the "fractional steps method" page 8 as well as cites the names of any Fortran files or similar.

The equation 1 is pretty alone in this context. Its presence is not necessary to deepen the results discussions. I think you can just refer to TEL3D documentation or previous works without described it. But if you prefer to include it you should also describe the whole equation system. In this case, I suggest to include them into an Appendix.

The table 1 is not clarifying too much the different nesting procedures. I suggest to enrich this table with the proper informations to clarify the different approaches.

Regarding the results, you have a lot of potential occasion for deepening the discussion which are missed. As a first, you should merge them into a unique Section (Model

[Figure]

Results) at least divided into 2 sub/sections. Than you have to improve them. For each different model accuracy analysis I suggest to quantitatively describe both the real measurements, by providing some statistics about the data (e.g. extremes and averages), and the differences with the model results, at least through RMSE computation or similar. The present version include, in most of cases, a qualitative description of the figures without any quantitative information.

I think the paper can be valorised also by a revision of the English language and grammar.

---

## Referee Comment (RC3) · Anonymous Referee #3 · 25 Apr 2016

* * *
General recommendations:
* * *
The present manuscript describes the implementation of embedded multiscale modelling systems in the Taranto Sea to refine predictions of hydrodynamics components (significant wave height, sea levels and currents) in the Mar Grande area. This research study provides, in particular, further insights about capabilities of a multiple-nesting method to improve numerical predictions at the local scale of harbor basin which requires generally improved temporal and spatial resolutions. Of particular interest is the availability of numerous in-situ measurements in the area considered. Nevertheless, strong differences are exhibited between observations and results from

the three-dimensional (3D) model in the Taranto Sea, particularly noticeable for depth-averaged velocities at the Mar Grande station. In addition, the manuscript focuses a bit too much on the comparison between predictions and observations neglecting a depth discussion about the hydrodynamics of the Mar Grande studied area. For these major two reasons, I suggest major corrections of the manuscript before publication in Natural Hazards and Earth System Sciences (NHESS). Improving the discussion will undeniably increase the potential of this very interesting research study. In order to help authors to review the manuscript which research deserves to be published in NHESS, major and minor comments are detailed in the two following sections.

———————

Major comments:

———————

1- On the basis of Figs. 11 and 12, it was concluded that the "combined effects of (i)...have led to a better representation of the circulation regime inside Mar Grande by the proposed modelling approach" in comparison with SANIFS predictions. But "this better representation" does not appear so well on these two figures. Indeed, the present modelling system predicts higher amplitudes of velocities than SANIFS reducing initial differences with in-situ measurements (Fig. 11). Nevertheless, strong differences still remain. In particular numerical results from TEL3D+TOM are found to overestimate observed velocities by 50 % on 6 October 2014. Taking into account these differences, it is very difficult to conclude that "the overall agreement of computations is quite good" here. What is more, whereas predictions from TEL3D+TOM provide better representation of daily-averaged vertical profiles of current at the Mar Grande station on 6 and 7 October 2014, a better agreement appears to be obtained with SANIFS on 5 October (Fig. 12). Extending the evaluation of both models predictions with additional measurements will confirm the better assessment of TEL3D+TOM in the studied area.

2- Whereas the evaluation of models predictions is of major interest in this research study, significant improvement of the quality of this manuscript will be gained by extending the discussion of results obtained here. For instance, numerical predictions may be exploited to gain further insights about the hydrodynamics of the Mar Grande area exhibiting major forcings and the dominant interactions including effects of currents on waves. Indeed, this latter aspect is not really discussed for the presentation of results in Fig. 6.
* * *
Minor comments:
* * *
p. for page and par. for paragraph

p. 1 – l. 26 = of tide and wave energy...

p. 2 – l. 3= You may add a reference about embedded models such as Guillou, Chapalain and Duvieilbourg (2013). Sea surface temperature modelling in the Sea of Iroise: assessment of boundary conditions. Ocean Dynamics, vol. 63, Issue 7, 849-863.

p. 2 – l. 7 = was furthermore the subject of numerous studies...

p. 2 – l. 10 = Holthuijsen, 2007

p. 2 – l. 10 = Are all the references cited compared predictions with in-situ observations? I am not sure of it. You may only include references which exhibit such comparisons (for instance, effects of currents on the significant wave height...).

p. 2 – l. 15 = temporal resolutions...

p. 2 – l. 19 = Guillou and Chapalain, 2012

p. 2 – l. 28 = in terms of wave forcings = only wave forcings ?

p. 3 = Please add references for TOMAWAC, TELEMAC2D and TELEMAC3D.

p. 3 = The last sentence has to be reviewed. It it not clear whether TELEMAC 3D is driven by predictions from TELEMAC2D or not.

p. 4 – l. 3 = to arrive to = to reach

p. 4 – l. 3 = The spin-up time does not really decrease with the increase in spatial and temporal resolutions but with the scale of the computational domain considered.

p. 4 – l. 12 = Please add a reference about the equation solved by TOMAWAC.

p. 4 – l. 30 = Do you consider effects of wave and current bottom boundary layers ? In particular effects of the apparent bottom roughness felt by the current above the wave boundary layer ? See Grant and Madsen (1979). You may add a comment about it here.

p. 5 – l. 28 = Nautical charts are often corrected to overestimate shallow waters areas. Is it the case in the present data?

p. 6 – l. 2 = I do not see the black circle in Fig. 2.

p. 6 – l. 22 = Not black circles but blue for locations of CTD measurements.

p. 7 - l. 4 = The Crotone station does not appear in Fig. 1.

p. 7 – l. 7 = What are the formulations retained for whitecapping dissipation and dissipation by bottom friction in TOMAWAC?

p. 7 – l. 8 = time step of COSMO wind data?

p. 7 – l. 18 = Wind drag coefficient and bottom friction appear to be corrected only for TEL2D and not for TOMAWAC. Please confirm it clearly in this paragraph.

p. 8 – l. 4 = well the tidal constituents...

p. 8 – l. 10 = What about the effects of currents induced-refraction on waves ? How do

you know that tidal flow induces steepening of waves here?

p. 8 – l. 24 = follows and following

p. 9 – eq. 1 = I will not display equation 1.

p. 9 = I will not specify the names of subroutines modified in TELEMAC.

---

## Author Comment (AC1) · 30 Jun 2016

Please find attached in the present, as separate *.pdf files in the compressed supplement, the document with the authors' response to the Referees' comments and the revision details, as well as the revised manuscript.

We remain at your disposal should you need any further information.

Respectfully, M.G. Gaeta, A.G. Samaras, I. Federico, R. Archetti, F. Maicu, and G. Lorenzetti

Please also note the supplement to this comment:
http://www.nat-hazards-earth-syst-sci-discuss.net/nhess-2016-95/nhess-2016-95-AC1-supplement.zip